# Position: Democratic AI is Possible. The Democracy Levels Framework Shows How It Might Work.

Aviv Ovadya[1]   Kyle Redman[1 2]   Luke Thorburn[1 3]   Quan Ze Chen[1 4]   Oliver Smith[1]
Flynn Devine[5]   Andrew Konya[6 1]   Smitha Milli[7]   Manon Revel[7]   K. J. Kevin Feng[4]   Amy X. Zhang[4]
Bilva Chandra   Michiel A. Bakker[8]   Atoosa Kasirzadeh[9]

## Abstract

This position paper argues that effectively "democratizing AI" requires democratic governance and alignment of AI, and that this is particularly valuable for decisions with systemic societal impacts. Initial steps—such as Meta's *Community Forums* and Anthropic's *Collective Constitutional AI*—have illustrated a promising direction, where democratic processes could be used to meaningfully improve public involvement and trust in critical decisions. To more concretely explore what increasingly democratic AI might look like, we provide a "Democracy Levels" framework and associated tools that: (i) define milestones toward meaningfully democratic AI—which is also crucial for substantively pluralistic, human-centered, participatory, and public-interest AI, (ii) can help guide organizations seeking to increase the legitimacy of their decisions on difficult AI governance and alignment questions, and (iii) support the evaluation of such efforts.[1]

## 1. Introduction

Who should steer the development of AI? Similar questions have emerged with previous technological advances (Ziewitz & Brown, 2013; DeNardis, 2014; Radu, 2019), and existing institutions and power structures will clearly play a significant role in adjudicating these questions. However, with AI's general-purpose nature, the pace of change, market

incentives, geopolitical incentives, and jurisdictional arbitrage opportunities pose unprecedented challenges (Allen & Weyl, 2024).

Thankfully, recent innovations in collective decision-making point towards a new generation of processes, infrastructures, and institutions to navigate these challenges (Ovadya, 2023a; OECD, 2020; CIP, 2024; Stilgoe, 2024). They provide new ways to ensure that the development of AI remains human-centered, not *just* at an individual human level, but societally and even globally; and not *just* by individual governments, but also by democratic processes commissioned by corporations, governments, and multilateral institutions. **Building on such democratic innovations, our position is that democratic AI is possible—and valuable for navigating the systemic societal impacts of AI.**

To illustrate more concretely what democratizing AI might look like, we provide the "Democracy Levels" framework which: (a) **articulates milestones** for the democratic AI (CIP, 2024), pluralistic AI (Sorensen et al., 2024; Kasirzadeh, 2024), participatory AI (Delgado et al., 2023; Groves et al., 2023; Wong et al., 2022), and public AI (Vincent et al., 2023; Public AI Network, 2024) ecosystems—a rapidly evolving set of organizations, institutions, and initiatives interested in ensuring that we have the necessary "democratic infrastructure" for navigating the transition to a world with highly-capable AI systems; (b) **can help guide organizations** and institutions that need to increase the legitimacy of difficult AI governance and alignment decisions; and (c) **supports evaluation** to identify opportunities for improvement of democratic systems and keep AI organizations accountable.

We see this framework as applicable to each of the yellow components in Figure 1: AI systems, AI organizations, and AI regulators (and the decision-making processes that feed into these). The ultimate intent is to provide a clear map of what it would take to enable meaningful democratic governance and alignment of AI, in a way that is useful both internally to organizations making decisions about AI, and externally to those supporting this work and providing accountability.

---

[1]AI & Democracy Foundation [2]newDemocracy Foundation [3]King's College London [4]University of Washington [5]Boundary Object Studio [6]Remesh [7]Meta AI, FAIR Labs [8]Massachusetts Institute of Technology [9]Carnegie Mellon University. Correspondence to: Aviv Ovadya <aviv@aidemocracyfoundation.org>.

*Proceedings of the 42nd International Conference on Machine Learning*, Vancouver, Canada. PMLR 267, 2025. Copyright 2025 by the author(s).

[1]This framework will continue to be evolved by the AI & Democracy Foundation (AIDF) and associated working groups at democracylevels.org.

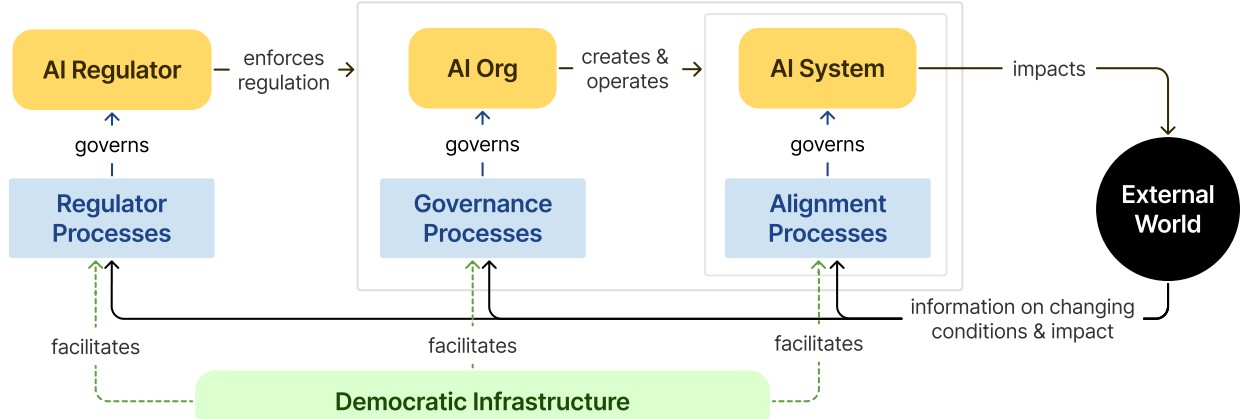

*Figure 1.* A system diagram of how democratic processes could integrate with the AI ecosystem, with democratic infrastructure being used to facilitate—where appropriate—collective decisions relating to AI regulation, organizational governance, and alignment. The Democracy Levels Framework we introduce can be used to evaluate (i) the degree to which democratic systems are used for decision-making, and (ii) the quality of those democratic systems, and the infrastructure supporting them.

## 2. Why Democratic AI?

Democracy and democratization are loaded terms, including in AI (Seger et al., 2023; Lin, 2024). We use *democracy* here to refer to processes and systems for collective decision-making that are "characterized by a kind of equality among the participants at an essential stage of the decision-making process" (Christiano & Bajaj, 2024), while acknowledging that democracy is a bundle of thick concepts that often include human rights, rule of law, institutional checks and balances, appropriate norms, sanctions, and other features that support such equality. *Democratic AI* then refers to AI systems whose development, alignment, and governance were substantively influenced or directly controlled by democratic systems. We emphasize that this use of democracy differs from how "democratization" is often used in the context of AI, where it is often understood as simply making AI open and accessible.

### 2.1. The societal challenge

Machine learning research is progressing at an unprecedented rate. AI systems now make explicit and implicit decisions affecting billions globally, from financial risk assessments (Sadok et al., 2022; Ramakrishnan et al., 2024) to recommender systems (Zhang et al., 2021; Deldjoo et al., 2024). We are increasingly subject to the decisions of (and interactions between) AI agents (Chan et al., 2023; Kolt, 2025; Kasirzadeh & Gabriel, 2025). Institutional entities and nation-states are rapidly developing AI systems designed to supplant human capabilities while amassing compute and energy infrastructure. This technological trajectory introduces several concerning possibilities: significant power concentration, difficult-to-govern proliferation of advanced capabilities to malicious actors (Allen & Weyl,

2024), or gradual disempowerment of most people (Kulveit et al., 2025; Drago & Laine, 2025). AI systems and those that control them may accrue unprecedented power. Who should wield that power? How should it be wielded?

### 2.2. Why democracy?

We face many consequential decisions about societal adaptation to AI (Section 6.1), and significant changes to the dynamics and distribution of power. In this context, democracy can be viewed as a (social) technology for responsibly making such decisions and managing such power shifts. More precisely, democracy is commonly justified as a way to *(i) legitimize the use of power* (e.g., via open and free deliberation, collective self-government, public justification, etc.; (e.g., Habermas, 1962; 1989; 1992; Cohen, 2002; 2003; Rousseau, 1762; Cordelli, 2020; Lafont, 2019; Young, 2002)), *(ii) distribute power* in ways that reflect intrinsic moral equality (e.g., Dahl, 1956; 1989; Buchanan & Tullock, 1962) and prevent domination (e.g., Pettit, 1997; 2012), and *(iii) achieve epistemic benefits* that are more difficult when power is in the hands of a few, rather than many (e.g., Estlund, 2005; 2009; Landemore, 2013; 2020).

The legitimacy and buy-in that democratic processes can generate are increasingly valuable to unilateral actors involved in the development and governance of AI at corporations, governments, and multilateral NGOs (Stilgoe, 2024). For corporations in particular, we argue that, in addition to the benefits already listed, the costs of using democratic systems for several critical areas of decision-making are substantially lower than reactive compliance with regulation, forced reorganization under antitrust action, or loss of market value due to eroded public trust. Moreover, such democratic systems can help address core organizational

barriers such as a "bias for inaction" due to conflicting internal or external stakeholders. Such processes can also help handle controversial decisions, such as those related to sensitive cultural issues, tradeoffs between liberty and safety, or influential default behaviors of AI systems (e.g., Raghavan, 2024; OpenAI, 2025).

Finally, we believe that democracy is an asymmetric enabler—that on the margin, greater adoption of democratic processes and systems would relatively advantage good actors and outcomes over bad. In particular, democratic processes can facilitate a number of political "moves" that may help mitigate risks of race dynamics and unstable multipolar regimes that could plausibly arise as AI technology advances. For example, democratic processes may help signal benign intent, demonstrate Pareto optimality, surface common ground, institutionalize conflict, provide stability by guarantees on process, and be a source of accountability by showing a difference between public will and unilateral actions.

## 2.3. How democratic AI can work

What could AI democratization look like? There are multiple possible approaches (Seger et al., 2023; Himmelreich, 2023; Project, 2023); pragmatically, however, the examples we discuss predominantly draw on contemporary deliberative democratic processes (Fishkin, 2011). This is due to their potential to simultaneously: work with jurisdictions of arbitrary size (including those with less identity infrastructure, and globally) (OECD, 2020) ; complement a wide variety of existing political structures (Fishkin et al., 2010) ; enable informed exchange incorporating diverse perspectives and expertise (Setälä et al., 2010; Landemore, 2012; Dryzek et al., 2019; Curato et al., 2021) ; and generatively identify common ground around charged issues (Ugarriza & Caluwaerts, 2014).

A democratic process of this form has as its input a **remit** and **constituent population**, and as its output a **decision**. The remit is a prompt that scopes the decision that needs to be made (and may specify the structure and properties required for the output). At an essential stage in the process, decisions are made by a representative subset of the constituent population (generally selected by sortition (OECD, 2020)). Processes are often conducted by a third-party democratic infrastructure provider (Ovadya, 2023a), roughly analogous to an election system vendor, who has expertise in conducting such deliberative processes (e.g., Involve; Nexus; MosaicLab; MASS LBP).

There are already early examples of democratic AI experimentation, including in companies. Most leading AI organizations have begun experimenting with democratic processes for policy or alignment decisions—including Anthropic's *Collective Constitutional AI* (Anthropic, 2023),

OpenAI's *Democratic inputs to AI* grant program (Eloundou & Lee, 2024), Meta's *Community Forums* (Broxmeyer, 2024), and Google DeepMind's STELA project (Bergman et al., 2024)—and there is increasing pressure to use such processes for the development of international regulation (Connected by Data, 2024; Ovadya, 2023a). Although these initial efforts are imperfect, they are helping build the organizational knowledge and capacity needed for more transformative outcomes. But to fulfill the potential of these processes and ensure they are credible, we need a shared language to describe and evaluate progress, which we will discuss in Sections 3 and 4.

## 3. The Democracy Levels Framework

Effective democratic AI requires compatibility between three key elements: the **systems** in place for democratic decision-making, the **level** of power granted to those systems, and the **context** in which decisions are made.

In this section, we introduce the core of the Democracy Levels Framework, which consists of (i) a set of Levels (Section 3.2) that capture the degree to which decision-making power has been transferred to a democratic system, and (ii) a set of dimensions (Section 3.3) that capture the qualities of that democratic system. Section 4 builds on this, with (iii) a Levels Decision Tool for informing decisions about "how much democracy" is appropriate in a given decision-making context, and (iv) a Democratic System Card that can be used to evaluate a democratic system against the dimensions.

### 3.1. Terminology

**Scope of Authority**: The set of *powers* that an authority is granted, including a specification of the *domain(s)* governed by that authority, and potentially implicit *external constraints*. For example, the scope of authority of a finance committee for a US corporation's board of directors is set out by a corporate charter, bylaws, and board resolutions; constrained by regulations and case law; and can have its authority limited to financial matters.

**Unilateral authority**: An entity that can make decisions without meaningful checks on its power within a given scope of authority (i.e., no approval needed for any decisions within its scope).

**Democratic Process**: A collective decision-making process characterized by "a kind of equality among the participants at an essential stage of the decision-making process" (Christiano & Bajaj, 2024).

**Democratic System**: A set of interacting entities and democratic processes (e.g., via checks and balances).

| | Roles Performed by Democratic System(s) | Description | Example |
|---|---|---|---|
| **L0** | ∅ | **Unilateral** decision-making: all formal decision-making authority lies with the unilateral authority. | *Rules on AI persuasion are simply created by the unilateral authority.* |
| **L1** | ⓘ | Outputs of a democratic process **inform** the unilateral authority; such democratic processes are initiated ad-hoc when desired and with a remit chosen by the unilateral authority. | *The process outputs recommendations on AI persuasion, which need to be interpreted by the unilateral authority for implementation as rules.* |
| **L2** | ⓘ ☰ | Democratic processes output a fully **specified** decision which must be implemented by default unless the unilateral authority uses a predetermined process or set of criteria to amend or veto. | *The process outputs rules on AI persuasion, which are implemented as-is, unless amended or vetoed.* |
| **L3** | ⓘ ☰ 🔨 | Democratic process outputs are **binding** and cannot be vetoed (assuming feasibility, e.g., technically, legally; and within their remit). | *The process outputs rules on AI persuasion, which are implemented as-is (unless a pre-established process finds it infeasible).* |
| **L4** | ⓘ ☰ 🔨 🏁 | The unilateral authority pre-commits to the automatic **initiation** of binding democratic processes when a given condition is met (instead of being initiated ad-hoc), with scope over a pre-specified domain. | *Processes to update rules on AI persuasion are run yearly or whenever a newly pretrained model is to be deployed.* |
| **L5** | ⓘ ☰ 🔨 🏁 🏛 | The unilateral authority fully shifts power within a domain of decision-making to an adaptive "constitutional order" — a system of checks and balances capable of making **metagovernance** decisions about when and how democratic processes are to be used (within a domain). | *The decisions around when to trigger processes to update rules (and how those processes are triggered) are also under the control of democratic processes (via a system of checks and balances).* |

| ⓘ informing decisions | ☰ specifying options | 🔨 making decisions | 🏁 initiating processes | 🏛 metagovernance |
|---|---|---|---|---|

*Figure 2.* **Overview of the Democracy Levels** (names of each in bold), which are used to assess how much decision-making power in a given domain of decision-making has been transferred from a unilateral authority to a democratic process. The example column describes (hypothetical) democratic systems operating at each level, with the scope of authority being rules around the use of AI systems for persuasion. Evaluating the ultimate democraticness (beyond power transfer) of a system must also consider the dimensions (Section 3).

Under these definitions, elections, citizen assemblies (OECD, 2020), and collective dialogue processes (Konya et al., 2023; Ovadya, 2023c) are *democratic processes*. The interactions between those processes and *unilateral authorities*, constituents, stakeholders, media, etc. make up a *democratic system* (supported via *democratic infrastructure*). A more complex democratic system might involve an entire constitutional order with multiple institutions interacting through different processes on an ongoing basis. For example, a repeated collective alignment process, feeding into a model spec (OpenAI, 2024b), managed by a democratic oversight body, all coordinated by democratic infrastructure providers, could have a *scope of authority* over AI model alignment.

### 3.2. Levels

The transfer of decision-making power from a unilateral authority to democratic systems can take many forms, but there are discrete points of particular significance that may require new democratic infrastructure and other structural changes. We developed the Democracy Levels to provide clearer distinctions for understanding and implementing these transfers, building on experience supporting movement between these levels.

We define each level of democratic decision-making according to which of five roles is performed by democratic systems, rather than a unilateral authority: (i) **informing** decisions; (ii) **specifying** decisions; (iii) making **binding** decisions; (iv) automatically triggered **initiation** of binding decision-making processes; and (v) **metagovernance**. Figure 2 provides definitions for each level, from L0 to L5, along with concrete examples of what this could look like in practice for a plausible decision domain: developing a set of rules governing persuasion by an AI system. Such rules might be used directly in model training (e.g., to align an AI system) (Mu et al., 2024) or as policies (e.g., for an AI

organization or regulator).

At L1 the democratic authority only provides input, which is interpreted as the unilateral authority wishes, and thus can be in any form. At L2 the decisions need to be directly implementable (e.g., a policy, model spec, etc.), but may be vetoed by a predetermined process. With L3 onward, all decisions are binding (within their scope of authority). L4 involves a democratic system that automatically initiates L3 binding processes when particular conditions are met. Finally, at L5, the democratic system also does metagovernance within its scope of authority (e.g., making decisions on how to run L4 processes), potentially via a set of checks and balances across different bodies and processes.

This framework particularly takes inspiration from the autonomy levels defined for self-driving cars (SAE International, 2021), which also involve the shifting of power and responsibility from a unilateral authority (i.e., human driver or AI corporation) to a new kind of decision-making system (i.e., autonomous control system or democratic system).

A single AI organization, government, or AI system may simultaneously implement multiple democratic systems, at different levels, for different decision domains. For example, an AI organization's decisions about whether to release a new model might be at L2, while decisions about the model spec used for fine-tuning could be at L4. There is also flexibility on the scope of authority for a process—for example, it is possible to have an L3 process with the scope of authority specified as binding for two years, or until a given condition is met (such as a model passing a particular benchmark).

### 3.3. Dimensions

While the *levels* specify which roles are performed by democratic systems (versus the unilateral authority), the *dimensions* (Table 1) convey the extent to which democratic systems are "good enough" to support the meaningful, safe, and effective implementation of higher democracy levels.

In some cases, if a democratic system is deficient in some way (such as leaving participants uninformed, lacking representativeness, or not being robust to adversaries), blindly moving to higher democracy levels can be risky since it could result in binding to a poorly made decision. The assessment of dimensions is also context sensitive: whether a democratic system is "good enough" can depend on specific contextual factors like the public's trust, understanding of the process, and willingness to participate, etc.

Below we describe three primary *dimension* categories—process quality, delegation, and trust—each with several dimensions.

For a democratic system or infrastructure component to be

able to support more decision-making responsibility, it must be able to reliably provide a certain level of **process quality**. More concretely, process quality dimensions evaluate whether the decisions produced are *representative* of the relevant population, participants are *informed* on the issue, processes involve *deliberative* reasoning, decision outputs are *substantive*, systems are *robust* to adversarial behavior and less-than-ideal conditions, and the final traces are *legible* (transparent and understandable) to non-participants.

Additionally, the unilateral authority needs to be able to effectively **delegate** to the democratic system. This includes the capacity of the unilateral authority to organizationally and publicly *commit* to the outcomes; to *integrate* such processes into its operations; and to technically and/or legally *bind* itself to the resulting decisions.

Finally, to ensure the process decision is accepted, there must be external conditions that support the success of the process, which we collectively refer to as **trust**. Specifically, the relevant public and stakeholders must be sufficiently *aware* of the process, buy into its *legitimacy*, be willing to *participate*; and there must be sufficiently capable forms of *accountability*.

### 3.4. Example Application

To further illustrate our framework, we can apply the levels to discuss the transition of decision-making power in the wild. For example, Anthropic's Collective Constitutional AI (Anthropic, 2023) effort involved a roughly representative microcosm of the United States public providing and evaluating principles for an AI system to follow. These principles were de-duplicated and slightly transformed for training a research model (Anthropic, 2023), with one of the clauses used for training a *deployed* model (Anthropic, 2025). This process informed model development, so it can arguably be seen as an example of a transition from L0 (unilateral) to L1 (informing decisions). Had Anthropic predetermined a process or criteria for de-duplicating, transforming, and ultimately accepting or rejecting the output of the process, this could have been an example of an L2 process. With a *binding* commitment to adopt, the same process could even be brought up to L3.

Another example can be found in Meta's Oversight Board's content moderation decisions about individual posts, which corresponds to a transfer of decision-making power at L4 (regular binding decisions). The Oversight Board's broader *policy advisory opinions* are *non-binding* and thus only operate at L1 (Meta Oversight Board, 2024). All that said, evaluating the dimensions, we can observe that the Oversight Board was not designed to be democratically representative. In contrast, Meta's Community Forums on AI are intended to be representative; however, they only *inform* policies and product decisions, and so are also L1 (Chang et al., 2024).

### 3.5. Related Frameworks

To develop this framework, we have drawn inspiration from existing frameworks for evaluating democratic-ness (Arnstein, 1969; Lindberg et al., 2014; IAP2 Australasia, 2024; Skaaning & Hudson, 2023), as well as frameworks for evaluating degrees of responsible behavior and autonomy in AI systems (Bommasani et al., 2024; SAE International, 2021). Our work relates to explorations and assessments of democratic (CIP, 2024), participatory (Delgado et al., 2023; Cooper & Zafiroglu, 2024; Suresh et al., 2024), pluralistic (Sorensen et al., 2024), human-centered (Sigfrids et al., 2023), and public AI (Public AI Network, 2024; Vincent et al., 2023). We discuss more details of these efforts surrounding democratic AI in the Appendix F.

## 4. Democracy Levels Tools

While the main components of levels and dimensions offer a common language to discuss the allocation and transition of decision-making power, in practice, balancing key elements of democratic AI (context, level, system) when planning for transition can be challenging. To make it easier for stakeholders to plan and evaluate possible transitions between levels, we introduce two tools that are meant to help ground the thinking process for various stakeholders: the *Levels Decision Tool*, intended to support unilateral authorities (and advocates) in planning out the transition to higher levels of democratic decision-making; and the *Democratic System Card*, intended to support the assessment of democratic systems so as to understand their fitness for adoption at a desired level and to identify opportunities for innovation and improvement.

### 4.1. The Levels Decision Tool

The *Levels Decision Tool*, a version of which is provided in Appendix B, was developed to pragmatically help evaluate the potential reasons for delegating a decision or decision-domain to a democratic system. It may be used by unilateral authorities directly for their own decision-making, or by stakeholders and advocates seeking to identify the appropriate arguments needed to demonstrate the value of democratic systems to those authorities.

Which level to aim for depends on the **context** of the decision, including: the unilateral authority; its scope of authority; who is affected by the decision; how the authority relates to other stakeholders, both internal and external; etc.

The Levels Decision Tool contains a set of targeted questions around this context that can help decision-makers consider which democracy level to aim for. The questions involve: the value of legitimacy across the public, internal stakeholders, external stakeholders, and government; the potential benefits of collective intelligence; the feasibility of transfer-

ring decision-making power; the importance of speed and adaptability to the decision domain; resourcing; and novelty. Some questions are applicable to every context, and some are only applicable to, e.g., corporations.

### 4.2. The Democratic System Card

The *Democratic System Card* is intended to help decision-makers document, assess, compare, and evaluate democratic systems in a structured manner, with a core goal of providing insight into how appropriate a system is at a given democracy level, for a given context. The system card (Table 1 and Appendix C) also provides an elaboration of the three *dimensions* we derived in Section 3.3: process quality, delegation, and trust.

A full system card has three primary components: (1) **descriptions** of how the democratic system works overall with respect to each dimension; (2) **assessments** of the system implementation with respect to each dimension, based on a series of *guiding questions*; finally (3) a qualitative **evaluation** of the highest level of power that the system can be trusted with for making decisions (for a given context).

In a similar vein to model cards and AI system cards (Mitchell et al., 2019; Anthropic, 2025; OpenAI, 2024a), democratic system cards support evaluating a democratic system within different contexts. Decision-makers can use system cards to assess whether a democratic system is ready to be delegated to higher levels of decision-making power, with more clarity around which dimensions within the system are the current bottlenecks. Stakeholders and advocacy groups can make use of system cards to compare and contrast possible alternative systems to propose or advocate for. While active and empowered democratic systems (or proposals for complete democratic systems) should have complete cards, prototypes, and research projects may have only parts of the card filled out, as not every aspect is applicable.

Democratic system cards are also meant to be a resource for those identifying gaps in the democratic infrastructure ecosystem, including those exploring potential opportunities and applications of their research. They can be used by process designers and democratic infrastructure providers for understanding critical needs for operating at higher levels of delegated power, and to consider which combinations of democratic processes complement one another.

## 5. Alternative Views

Below, we summarize common objections and responses to them in question-and-answer format. Additional alternative views are listed in Appendix E.

*Q. Shouldn't people make their own choices about how they use AI?* This **libertarian critique** argues that individuals

*Table 1.* **Democracy Level Dimensions** and **Democratic System Card** overview—The dimensions of the Democracy Levels Framework and the corresponding democratic system card questions. These are used to guide reflection on whether the quality of a democratic system is commensurate with the level of decision-making power delegated to it. See Appendix C and democracylevels.org/system-card for details, examples, and templates for filling out Democratic System Cards.

## Process Quality *(relates to the democratic infrastructure)*

| | *The extent to which ...* | |
|---|---|---|
| Representation | key decisions are representative of the constituent population. | To what extent: **1** is there sufficient representation at critical parts of the process, including (a) proposing decisions, and (b) making ultimate decisions? **2** are there barriers leading to bias in representation? |
| Informedness | critical information is taken into account for decision-making. | To what extent: **1** is critical context incorporated into decision-making about tradeoffs and consequences of different decisions? **2** is this sourced from (a) domain expertise, (b) the existing authorities, who may have extensive context, (c) a broad diversity of constituents, (d) the most impacted stakeholders, and (e) the powerful stakeholders, whose incentives are critical to having the decision "stick"? |
| Deliberation | decisions are considered and deliberative (rather than superficial and reactive). | To what extent are those involved: **1** able to (and supported to) move from shallower to deeper goals and values? **2** able to (and supported to) collaborate where necessary? **3** able to address issues within the available time? |
| Substantiveness | decisions are substantive (e.g., actionable, consequential) rather than nonsubstantive (e.g., vague, simplistic, inconsequential). | To what extent: **1** is the decision directly actionable and implementable? **2** does the decision meaningfully address the issues? **3** does the decision grapple with the necessary levels of complexity? **4** is uncertainty appropriately managed and accounted for? **5** are risks to implementability accounted for? |
| Robustness | the process is robust to suboptimal conditions or adversarial or strategic behavior. | To what extent is the process or system vulnerable to: **1** suboptimal conditions or broken assumptions? (e.g., low turnout, larger power asymmetries) **2** strategic behavior and manipulation? **3** false claims? (e.g., of manipulation) |
| Legibility | the processes and decisions are accessible, understandable, and verifiable | To what extent is information (a) accessible, (b) understandable, (c) verifiable about the: **1** processes/systems used to make decisions? **2** the execution of these processes? **3** decisions being made **4** reasons and inputs feeding into decisions? |

## Delegation *(relates to the unilateral or existing authorities)*

| | | |
|---|---|---|
| Integration | the authority integrates the democratic process into its operations. | To what extent is the authority structuring its internal communications and operations to effectively: **1** provide critical context to democratic process / system? **2** integrate democratic process outputs in its actions? **3** trigger democratic processes when/if required? |
| Ability to bind | the authority is able to technically and legally bind itself to democratic decisions. | To what extent can the unilateral authority bind itself to acting in accordance with the democratic decision: **1** technically? **2** legally? (e.g., has developed the needed technical and/or legal infrastructure for binding) |
| Commitment | the unilateral authority commits to acting in accordance with the democratic decision. | To what extent has the unilateral authority committed to acting in accordance with the democratic decision: **1** internally? **2** privately? **3** publicly? (regardless of their ability to bind) |

## Trust *(relates to the external conditions)*

| | | |
|---|---|---|
| Awareness | the relevant public is aware of the democratic process. | To what extent is the relevant public aware: **1** that the democratic system exists? **2** how it works? **3** what it is being used for? **4** how they can be involved? |
| Participation | the relevant public is willing to participate in the process. | To what extent is the relevant public: **1** willing to participate? **2** able to participate? **3** appropriately compensated for participating? **4** actually participating? |
| Accountability | there are external watchdogs and accountability structures monitoring the execution of the democratic process and the implementation of its outputs. | To what extent are: **1** there well understood lines of oversight and accountability? **2** sufficiently influential/powerful organizations focused on holding authorities to their promised levels of democratic involvement? **3** authorities and democratic systems responsive to such accountability mechanisms? |
| Buy-in | the relevant public and key stakeholders buy-in to the process and its legitimacy. | To what extent are the relevant public and key stakeholders accepting of the legitimacy of: **1** the system/process? **2** of the decision? |

are the best judges of what is in their interests and should be free to make their own choices, provided that those choices do not harm others (Locke, 1887; Mill, 1966).

*A.* We agree—with critical caveats. Most decisions would ideally be made by individuals/users; however, democratic processes may be needed for addressing systemic impacts, significant externalities, and decisions where revealed preferences diverge significantly from more reflective or deliberative preferences. Even without these conditions, there often remains a significant barrier to individual choice in practice, given the lack of interoperability and thus friction to move between different AI systems (particularly when they are directly integrated into devices, products, and services).

*Q. Shouldn't governments just regulate?* This **government-only critique** cites corporations' poor track record of addressing the systemic issues relating to their activities.

*A.* While government action is crucial (and the Democracy Levels Framework can be applied to regulators) it is insufficient alone. Given the potentially extraordinary benefits and risks of AI, and the influence of corporations on the trajectory of AI development, a full-stack approach to democratic AI is likely to be necessary. The organizations developing AI systems are closer to the critical context needed to make informed decisions and can make decisions more rapidly than governments can respond with regulation (Brookings Institution, 2023). The framework can help to avoid democratic legitimacy relying on the slowest-moving actor in a jurisdiction.

Governments may also themselves be less democratic, with both autocrats and politicians seeking to concentrate and entrench their own power. It may be much faster to innovate on creating fit-for-purpose democratic systems for governing AI by experimenting in industry, and then bringing them into government once the systems are more mature. Finally, the organizations developing AI, the use of AI, and the risks of AI can all be transnational—beyond the jurisdiction of any single government.

*Q. Shouldn't AI corporations focus solely on shareholders and profit?* This **shareholder-first critique** argues that shifting power from shareholders and executives to democratic systems would decrease societal benefit, under the assumption that maximizing corporate profits for shareholders will also maximize societal good (Friedman, 1970).

*A.* Shareholder maximization may be societally beneficial—if externalities are sufficiently internalized—but this generally requires effective government action (see previous answer). The competing Stakeholder Theory aims to address some of these gaps (Freeman, 1984; Mahajan et al., 2023) by encouraging corporations to take into account stakeholder impacts—and democratic processes are precisely a means for implementation. Moreover, most shareholder pressure is

also a fairly slow and blunt instrument, with significant lag; given the pace of AI change, a more responsive form of societal feedback may be critical. Broad-based societal benefits seem especially unlikely if profit-maximizing AI-first corporations continue to increasingly dominate the economy, leaving more and more people outside of both democratic and economic feedback loops (Kulveit et al., 2025).

That said, even under a shareholder primacy model, trust and legitimacy may become increasingly salient for corporate executives and shareholders as the societal import of AI increases, due to a mix of procurement, regulatory, and universal owner pressures (Hawley & Williams, 2000; Mattison et al., 2011; Docherty, 2024). As the Levels Decision Tool (Section 4.1) shows, there are a number of potential benefits from devolving some power.

*Q. Wouldn't this slow down the development of AI?* This **accelerationist critique** reflects a concern that the use of democratic processes will stymie AI progress and development, and might reduce the capabilities of democratic nations compared to their non-democratic counterparts, negatively impacting valuable innovation and the relative power of democratic nations (U.S.-China Economic and Security Review Commission, 2024; Andreessen, 2023).

*A.* Democratic systems *can* be very slow and ineffective, but that is not inevitable, especially with sufficient investment in democratic innovation, such as the development of augmented deliberative democratic processes. Concern about the relative position of democratic versus non-democratic states is driven, at least in theory, by a desire to protect democratic values—and applying democratic processes to AI governance is a demonstration of exactly those democratic values and their benefits. High-quality decision-making and regulation from improved democratic processes may even create conditions that accelerate innovation (Bradford, 2024).

## 6. Discussion

Given the increasingly systemic impacts of AI, it seems increasingly plausible that democratic AI will be a necessary (though insufficient) part of any potential positive future. The machine learning community has a significant role to play in enabling that future, both by improving our democratic infrastructure and by helping raise awareness about the importance of connecting technical capabilities to democratic oversight, particularly as AI systems become more powerful.

### 6.1. What decisions should be made democratically?

The kinds of decisions that should be made democratically are contestable, and while the Levels Decision Tool (Section 4.1) summarizes relevant considerations, our frame-

work is intentionally agnostic on this question. However, in broad strokes, we think there is a stronger moral case for a unilateral authority to devolve decision-making power to a democratic process or system when (a) the decision involves externalities (costs or benefits to third parties that are not borne by the unilateral authority), (b) those impacted by the decision cannot easily or reasonably opt out of experiencing such impacts, and (c) the impacts of the decisions are substantive enough to make up for the costs of the democratic system. Without intending to be stipulative or exhaustive, we think examples of decisions that may plausibly fit these criteria include:

- Governance decisions made by **AI regulators** that significantly alter societal norms or socioeconomic conditions (e.g., *guardrails/limits on human-AI relationships; whether/how AI agents can be legal persons; whether/how AI agents can participate in the economy; whether/how AI agents can replace human workers*).

- Development and deployment decisions made by **AI organizations** that impact large user bases, have significant cascading impacts, pose significant systemic/societal risks, or significantly accelerate or shift societal trajectories relative to counterfactual alternative decisions (e.g., *decisions around what safety thresholds must be met before release/deployment of increasingly capable models; decisions to significantly alter the behavior of models to which many people have formed economic or emotional dependencies; decisions relating to what organizational structures and incentives influential labs subject themselves to*).

- Decisions made by **AI systems** for which there should be "society-in-the-loop" (Rahwan, 2018; Konya et al., 2023) (e.g., *decisions made by AI systems that play significant roles in the functioning of utility-scale infrastructure*).

### 6.2. Future Directions

There is a vast array of critical work needed to make democratic AI a reality across AI systems, regulators, and organizations—involving a mix of evaluation, system design, pilots, and institutionalization. As a concrete technical example, social choice analysis can help us understand the extent to which the current machine learning paradigms may already be implicitly democratic or undemocratic (Conitzer et al., 2024).

Advancements in AI may also help improve many kinds of democratic processes and systems, with recent studies demonstrating their potential to mediate human deliberation and find common ground (Bakker et al., 2022; Fish et al., 2024; Summerfield et al., 2024; Tessler et al., 2024; Konya

et al., 2023; Goldberg et al., 2024). AI systems have also been used to synthesize diverse viewpoints within large populations whilst maintaining cultural and contextual nuances (Leibo et al., 2024; Bergman et al., 2024); and promote constructive disagreement (Summerfield et al., 2024; Burton et al., 2024). Applying these advances to build, test, and validate fit-for-purpose democratic systems can help increase the extent to which such systems are viable alternatives to unilateral decision-making.

To rigorously assess the "democratic-ness" of such AI-enabled democratic systems, research is needed to develop domain-specific metrics and evaluation frameworks. Both approaches—"democracy for AI" and "AI for democracy"—would benefit from further research on integration into existing governance and technical structures and processes (Reuel et al., 2024).

### 6.3. Conclusion

Maturity in the democratic governance of AI won't come overnight—organizations, democratic infrastructure providers, stakeholders, and the public all need to build democratic muscle—and taking on too much all at once can backfire (Smith, 2009). Instead of holding organizations to a platonic ideal, it can often be more helpful to focus on improvements relative to the status quo or comparable organizations, both of which can be articulated through a leveling and evaluation system. Spelling out the larger milestones toward achieving an audacious goal can also provide motivation to achieve those goals, and even enable the creation of powerful incentives like advanced market commitments (Kremer et al., 2020).

The framework that we provide can be used to build up a democratic capacity and to clarify when organizations are claiming to be acting more democratically than they actually are. This can then provide a basis for ensuring that they live up to their professed standards. Such differentials between ambitious democratic aspirations and reality have been a major force for democracy across history (Dahl, 1989). True democratization is a journey, and we aim to have provided a useful map.

AI is reshaping the world. If we are to preserve liberty and agency over our future, we see few alternatives: *we must have democratic systems fit for the age of AI*. By providing a concrete articulation that may be contested and built upon, we hope that the call for democratic AI, and the Democracy Levels Framework, may enable more productive conversations about what futures we should be aiming for with regard to power, participation, pluralism, and democracy.

## Acknowledgments

The authors would like to thank Danielle Allen, Nora Ammann, Liz Barry, Joe Edelman, Joseph Gubbels, David Evan Harris, Maximilian Kroner Dale, Sarah Hubbard, Jonas Kgomo, Seth Lazar, Lorenzo Manuali, Evan Shapiro, Anka Reuel, Allison Stanger, Jessica Yu, Glen Weyl, Kinney Zalesne, and others for feedback on earlier versions of this paper. Any remaining limitations are those of the authors.

Luke Thorburn was supported in part by UK Research and Innovation [grant number EP/S023356/1], in the UKRI Centre for Doctoral Training in Safe and Trusted Artificial Intelligence (safeandtrustedai.org), King's College London.

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

# Appendices

## A. Democracy Levels Tool Application Illustration

When considering transitions of power to a democratic authority, stakeholders need to balance three key elements: the **context** in which the decisions are made, the **level** to target the transition to, and the democratic **systems** that would be used. The Democracy Levels Tools are structured specifically to ground the thinking on these tasks, as illustrated in Figure 3: (1) the **Levels Decision Tool** helps ground thinking around what **level** to aim for given the *context* for a decision, which supports high-level planning ahead of potential transitions; (2) the **Democratic System Card** helps ground thinking around whether a specific democratic **system** is sufficient for the demands of the *context* and target *level*, which supports lower-level planning around the specific democratic system or process to adopt in a transition.

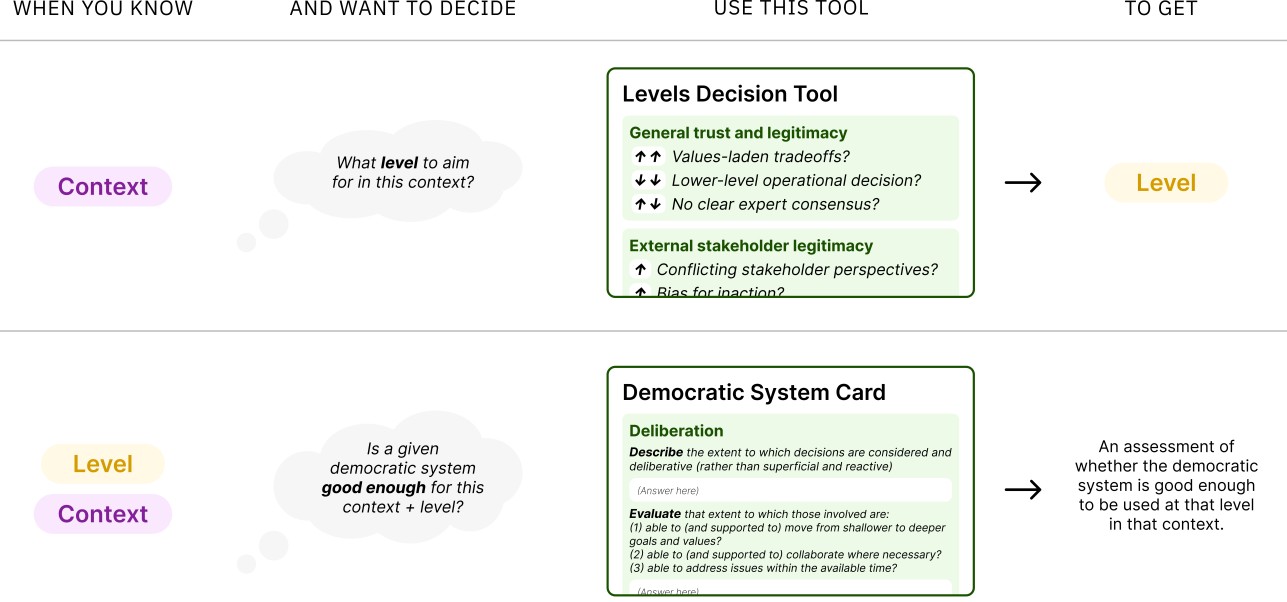

*Figure 3.* A diagram illustrating how the Levels Decision Tool and Democratic System Card can be used to assist stakeholders with planning the *transfer of decision-making power* and evaluation of *democratic systems to adopt* within their specific decision-making context.

## B. Levels Decision Tool

The Levels Decision Tool (Section 4.1) is a set of questions to help determine which Democracy Level to aim for in a given context. The arrows refer to whether each question, if answered in the affirmative, gives reason for targeting a higher or lower Democracy Level. In some cases, both arrows are used to indicate that the implication for what Level to aim for will be context-dependent.

You can find the most up-to-date version of the Levels Decision Tool, including an editable Google Docs version at democracylevels.org/decision-tool.

### General trust and legitimacy

*To what extent:*

- ↑↑ Is trust and legitimacy important to the unilateral authority?
- ↑↑ Is the unilateral authority not trusted by other actors, regardless of the decisions it makes?

*To what extent does the decision involve:*

- ↑↓ No clear expert consensus?
- ↑ Significant public interest concerns or externalities?
- ↑↓ Limited public impact?
- ↑↓ No clear expert consensus?
- ↓ A private technical or operational matter?

### External stakeholder legitimacy

*To what extent is there:*

- ↑ Powerful stakeholder groups, with conflicting perspectives?
- ↑ A bias for inaction given such conflict?
- ↑ An opportunity for decreased criticism or consequences due to "process legitimacy"

### Internal stakeholder legitimacy

*To what extent is there:*

- ↑ Significant internal disagreement?
- ↑ A need to evolve organizational values or purpose with a strong mandate?
- ↑ Talent motivated by public benefit?
- ↑ A history of collaborative decision-making?
- ↑↓ Hierarchical structure
    - ↑ Can make it easier to delegate power to a democratic system
    - ↓ Can correspond to reduced respect for such systems

- ↑↓ Extensive collaboration between teams / departments?
    - ↑ Can correspond with flexible systems
    - ↓ Can indicate complex interdependencies

### Government risk

*To what extent is there:*

- ↑ Risk of backlash (e.g., antitrust, regulatory) if power is too concentrated?
- ↑ Pressure from politicians or autocrats to act in a way clearly against the wishes of the public?
- ↑ Limited oversight?
- ↑↓ High scrutiny environment?
    - ↑ Proactive democratization can accelerate preferential regulatory outcomes
    - ↓ Lack of regulator experience with democratic processes or systems may limit influence
- ↑↓ Complex regulatory landscape?
- ↓ Clear existing law?

### Collective intelligence

*To what extent does the decision:*

- ↑ Benefit from a broad diversity of perspectives?
- ↑ Benefit from locally distributed knowledge?
- ↑ Involve high levels of uncertainty?
- ↓ Involve complex interdependencies?

### Viability

*To what extent:*

- ↑↓ Are there political obstacles to delegating power?
    - ↑ Democratic legitimacy could resolve barriers
    - ↓ Political obstacles are despite democratic norms
- ↓ Are there organizational structure, legal, technical, or physical obstacles to further delegating or devolving power?

## Resources

*To what extent:*

- ↑ Is the level of resourcing made available for the democratic system commensurate with the importance of its decisions?
- ↑ Can resources be made available for recurring expenses (only impacts L4, L5)?

## Speed

*To what extent does the decision involve:*

- ↑↓ Time-critical responses?
  - ↑ Can establish processes suited to time constraints
  - ↑ L4 processes can respond to regular time-critical scenarios in specified ways
  - ↓ Rules out certain complex systems
- ↓ Emergency responses?

## Adaptability

*To what extent do you anticipate:*

- ↓ Rapid changes to internal or external conditions that might impact the decision, relevance, or remit of a process?
  - ↑ This may be counteracted by more repeating or adaptive L4 or L5 systems

## Novelty

*To what extent:*

- ↑↓ Is devolving such decisions novel?
  - ↑ Democratic first-mover might increase respect and newsworthiness
  - ↓ Exposed to unknown risks

## C. Democratic System Card

The Democratic System Card (Section 4.2) is a set of questions to guide reflection on whether the quality of a democratic system is commensurate with the level of decision-making power delegated to it, for a given domain of decision-making, in a given context. The questions are grouped by the dimensions in the Democracy Levels Framework.

To complete a System Card, you first:

1. Describe the process or system at a high level.

2. Summarize what other systems the process depends on or interacts with, which impact its success (e.g., sortition data, or user or citizen authentication systems).

3. Then, for each dimension/property defined on the card below, describe how that dimension works in the process or system you are evaluating.

4. Use the guiding questions to reflect on the extent to which that dimension is satisfied.

You can find the most up-to-date version of the Democratic System Card, including editable templates that can be initially filled with an LLM (given context) at: democracylevels.org/system-card.

---

## Context

**Describe:** *Describe the process or a system at a high level. (Can reference a process card for more details. Can call out what is unspecified.)*

(Answer here)

**Describe:** *What are other systems that this process depends on or interacts with, which impact its success? (e.g., sortition data, or user or citizen authentication systems)*

(Answer here)

## Process Quality

### Representation

**Describe:** *The extent to which key decisions are representative of the constituent population.*

(Answer here)

**Evaluate:** *To what extent: (1) is there sufficient representation at critical parts of the process, including (a) proposing decisions, and (b) making ultimate decisions? (2) are there barriers leading to bias in representation?*

(Answer here)

### Informedness

**Describe:** *The extent to which critical information is taken into account for decision-making.*

(Answer here)

**Evaluate:** *To what extent: (1) is critical context incorporated into decision-making about tradeoffs and consequences of different decisions? (2) is this sourced from (a) domain expertise, (b) the existing authorities, who may have extensive context, (c) a broad diversity of constituents, (d) the most impacted stakeholders, and (e) the powerful stakeholders, whose incentives are critical to having the decision "stick"?*

(Answer here)

### Deliberation

**Describe:** *The extent to which decisions are considered and deliberative (rather than superficial and reactive).*

(Answer here)

**Evaluate:** *To what extent are those involved: (1) able to (and supported to) move from shallower to deeper goals and values? (2) able to (and supported to) collaborate where necessary? (3) able to address issues within the available time?*

### Substantiveness

**Describe:** *The extent to which decisions are substantive (e.g., actionable, consequential) rather than nonsubstantive (e.g., vague, simplistic, inconsequential).*

(Answer here)

*Evaluate:* To what extent: (1) is the decision directly actionable and implementable? (2) does the decision meaningfully address the issues? (3) does the decision grapple with the necessary levels of complexity? (4) is uncertainty appropriately managed and accounted for? (5) are risks to implementability accounted for?

### Robustness

*Describe:* The extent to which the process is robust to sub-optimal conditions or adversarial or strategic behavior.

(Answer here)

*Evaluate:* To what extent is the process or system vulnerable to: (1) suboptimal conditions or broken assumptions? (e.g., low turnout, larger power asymmetries) (2) strategic behavior and manipulation? (3) false claims? (e.g., of manipulation)

(Answer here)

### Legibility

*Describe:* The extent to which the processes and decisions are accessible, understandable, and verifiable.

(Answer here)

*Evaluate:* To what extent is information (a) accessible, (b) understandable, (c) verifiable about the: (1) processes/systems used to make decisions? (2) the execution of these processes? (3) decisions being made (4) reasons and inputs feeding into decisions?

(Answer here)

## Delegation

### Integration

*Describe:* The extent to which the authority integrates the democratic process into its operations.

(Answer here)

*Evaluate:* To what extent is the authority structuring its internal communications and operations to effectively: (1) provide critical context to the democratic process / system? (2) integrate democratic process outputs in its actions? (3) trigger democratic processes when/if required?

(Answer here)

### Ability to bind

*Describe:* The extent to which the authority is able to technically and legally bind itself to democratic decisions.

(Answer here)

*Evaluate:* To what extent can the unilateral authority bind itself to acting in accordance with the democratic decision: (1) technically? (2) legally? (e.g., has developed the needed technical and/or legal infrastructure for binding)

(Answer here)

### Commitment

*Describe:* The extent to which the unilateral authority commits to acting in accordance with the democratic decision.

(Answer here)

*Evaluate:* To what extent has the unilateral authority committed to acting in accordance with the democratic decision: (1) internally? (2) privately? (3) publicly? (regardless of their ability to bind)

(Answer here)

## Trust

### Awareness

*Describe:* The extent to which the relevant public is aware of the democratic process.

(Answer here)

*Evaluate:* To what extent is the relevant public aware: (1) that the democratic system exists? (2) how it works? (3) what it is being used for? (4) how they can be involved?

(Answer here)

### Participation

*Describe:* The extent to which the relevant public is willing to participate in the process.

(Answer here)

*Evaluate:* To what extent is the relevant public: (1) willing to participate? (2) able to participate? (3) appropriately compensated for participating? (4) actually participating?

(Answer here)

### Accountability

***Describe:*** *The extent to which there are external watchdogs and accountability structures monitoring the execution of the democratic process and the implementation of its outputs.*

(Answer here)

***Evaluate:*** *To what extent are: (1) there well-understood lines of oversight and accountability? (2) sufficiently influential/powerful organizations focused on holding authorities to their promised levels of democratic involvement? (3) authorities and democratic systems responsive to such accountability mechanisms?*

(Answer here)

### Buy-in

***Describe:*** *The extent to which the relevant public and key stakeholders buy into the process and its legitimacy.*

(Answer here)

***Evaluate:*** *To what extent are the relevant public and key stakeholders accepting of the legitimacy of: (1) the system/process? (2) of the decision?*

(Answer here)

## D. Design Decisions

There are a few notable decisions embedded in this framework.

We intentionally divorce *process quality* from *delegation*, as third-party democratic infrastructure providers can be commissioned by unilateral authorities such as AI labs and regulators (Chang et al., 2024; Ovadya, 2023b; Broxmeyer, 2024; The Forum for Ethical AI, 2019; Anthropic, 2023). This separation of concerns can help prevent fraud and provide an opportunity for process improvements by one organization to be passed on to other organizations. Operationally, at L1 such commissioned deliberations are roughly analogous to commissioning representative surveys or community engagement processes, and above that level, the processes being commissioned are more sophisticated and more directly integrated into organizational decision-making.

These dimensions also have significant dependencies. For example, processes that fail to demonstrate sufficient process quality given the level of power entrusted to them (delegation) are likely to lead to backlash (low trust; e.g., if a process is subverted, or a democratic decision sounds good but ends up being counterproductive).

Most (if not all) of these dimensions have long been studied in democratic theory and related disciplines, and can be operationalized in different ways — see, e.g., the many conceptions of representation (Dovi, 2018). For now, we aim for the framework to be agnostic to these differences.

## E. More Alternative Views

*Q. Why should non-experts make decisions on technical matters?* This **technocratic critique** is concerned with poor decisions being made by non-experts in a democratic process because AI is highly complex and can be conceptually difficult to understand.

*A.* We agree that non-experts should not be making technical decisions in the same way that no one wants a non-medical professional to diagnose them. However, many of the decisions about AI, from which data to collect to goal specification, are or can be highly value-laden (Elish & Boyd, 2018). The assumption that such decisions are too complex is also debatable, given that there have been successful democratic processes, such as Citizens' Assemblies, that have involved non-expert members of the public to produce informed decisions on problems of similar complexity. These processes have mechanisms to deal with the expertise gap (via educational components, expert consultation, etc.) (Warren & Gastil, 2015; Leino et al., 2022).

*Q. Why would people want to take part in the democratic governance of AI?* The argument at the heart of this **busy lives critique** is that people are busy and so will not want to dedicate time to a task that they may see as a job for corporations and maybe the government.

*A.* We accept that not everyone will have the time or the interest to participate in the democratic governance of AI. However, the Democracy Levels Framework is compatible with a range of democratic processes; it is not the case that everyone needs to be interested all of the time for the democratic governance of AI to be viable.

*Q. Democracy isn't just collective decision-making or deliberation—what about [other important elements or theories of democracy]?* Some of these **democratic critiques** argue that there are other fundamental elements to democracy beyond mere preference aggregation, including respect for human rights, rule of law, equality before the law, and freedom of expression. Others might argue that the framework is tailored too closely to *participatory* ideals of democracy, and is less accommodating of (e.g.) *representative* (Urbinati, 2006) or *agonistic* (Mouffe, 1999) theories of what democracy is or should be.

*A.* We agree democracy is more than collective decision-making, especially when implemented at the level of nation states, but there are already frameworks for evaluating democracy in such settings (e.g., Lijphart (2012)). The core of our framework (the Levels) focuses strategically on the transfer of decision-making power from unilateral authorities to relevant communities because we believe it to be a necessary (if not always sufficient) element of "democratic-ness" that can be evaluated not just for governments and regulators but also for organizations and AI models. The framework is also agnostic with respect to the preferred theory of democracy; all theories describe some (formal or informal) process by which collective decisions are made, so they can be evaluated against the Levels (Section 3.2), and the desiderata described in the dimensions and Democratic System Card (Sections 3.3 and 4.2) can be applied and interpreted when using, e.g., representative or agonistic frames.

## F. Additional Related Work

Below, we expand on the related work described in Section 3.5, and describe how our framework relates to work on alignment, participatory AI, and human-centered AI.

**Alignment** Alignment procedures, such as RLHF (Ouyang et al., 2022) or Constitutional AI (Bai et al., 2022), generally aim to make AI systems more aligned with the preferences, values, or goals of humans. However, democratic considerations are often not central to these methods. For example, the annotators recruited for preference learning methods like RLHF or DPO (Rafailov et al., 2024) are often highly unrepresentative of the user population. More-

over, the standard versions of these methods assume that the collected preference comparisons come from one human, even when they actually come from many annotators who may have differing preferences (Siththaranjan et al., 2024).

Recently, there have been growing efforts to make alignment methods more democratic, including Anthropic's Collective Constitutional AI, which uses Pol.is to identify principles that an AI should follow from a representative sample of U.S. adults (Huang et al., 2024). As another example, Conitzer et al. (2024) called for explicitly applying social choice approaches to preference learning methods (Ge et al., 2024), instead of implicitly interpreting different annotators as one person. While these nascent efforts are exciting, there is as of yet no standard way to evaluate how democratic these new alignment approaches are—our framework aims to provide such guidelines.

**Participatory AI**  Another emerging approach that offers engagement with diverse voices in the design and deployment of AI is participatory AI. This encompasses a range of processes to engage people, from approaches that **consult** with stakeholders (e.g., expressing preferences for policies in a ranking), through to those in which stakeholders **own** the design process and play a central role in shaping the procedures for deliberation as well as deciding on the outcomes of the process (Delgado et al., 2023). Although this space shows promise, progress is nascent. In the private domain, efforts to date have largely been confined to consultation (Groves et al., 2023). In the public sphere, the implementation of participatory approaches faces difficulties in integrating with existing institutional rules and frameworks and agreeing on who should participate (Wong et al., 2022). Overall, it is felt by some that further clarity is required on the definition and role of Participatory AI, and how it relates to other available approaches (Birhane et al., 2022). We believe that the Democracy Levels Framework can help provide some of this clarity.

**Human-centered AI**  Our Democracy Levels Framework strives to make AI systems more human-centered. By *human-centered AI*, we mean AI systems that are developed by drawing upon human-centered design methods—in addition to purely algorithmic ones—to ensure that the AI can better serve human needs (Shneiderman, 2022; 2020; Riedl, 2019; Capel & Brereton, 2023; Auernhammer, 2020). Democratic AI and "traditional" human-centered AI share many goals; advancing one can often advance the other (Sigfrids et al., 2023). For example, democratic processes can involve iterative, deliberative decision-making from broad input, whereas traditional human-centered processes encourage iterative refinement through user and stakeholder feedback throughout the design lifecycle. Democratic processes empower individuals to collectively make gover-

nance and policy decisions that impact them, while human-centered design empowers users to shape systems in a way that aligns with their goals. As one moves up the Democracy Levels in our framework, constituents' agency is increasingly centered to provide them with more decision-making power over their AI systems.

## G. Example System Card: Habermas Machine

The following is an example partial system card for an AI tool meant to support collective deliberation (Tessler et al., 2024).

# Democratic System Card

## Context

*Describe: The process or a system at a high level. (Can reference a process card for more details. Can call out what is unspecified.)*

The Habermas Machine represents a novel approach to collective deliberation, leveraging LLMs to facilitate the discovery of common ground among individuals with diverse perspectives. Inspired by Jürgen Habermas's theory of communicative action, this AI system employs large language models (LLMs) to process personal opinions and critiques submitted by participants. Through iterative generation and refinement, the Habermas Machine produces group statements that aim to maximize collective endorsement and reflect shared understanding.

At its core, the system combines two fine-tuned LLMs: a generative model that proposes high-quality candidate group statements, and a personalized reward model (PRM) that evaluates these candidates based on each participant's personal opinion. The PRM outputs a personalized ranking of candidate statements for every participant. These rankings are then aggregated using a social choice function to select a group statement that aims to reflect broad agreement while incorporating critical minority viewpoints, facilitating inclusive and balanced consensus.

*Describe: What are other systems that this process depends on or interacts with, which impact its success? (e.g., sortition data, or user or citizen authentication systems)*

**User Input Quality:** The quality of the initial opinions and critiques participants provide is crucial. If users are not prepared to contribute in good faith or lack relevant information, the HM's output will be less effective.

**LLM Capabilities:** The success of the HM relies heavily on the capabilities of the underlying LLM to understand nuanced opinions, generate coherent and relevant group statements, and incorporate critiques effectively. Biases in the LLM could also impact the process.

**Platform for Interaction:** A reliable and accessible platform is needed for participants to interact with the HM, submit opinions and critiques, and receive group statements.

**Human Oversight:** While AI-mediated, human oversight is still important to ensure the process is fair, address any unforeseen issues, and interpret the outputs in a real-world context.

**Context of Deliberation:** The specific social or political issue being deliberated and the broader context of the deliberation can influence the effectiveness and impact of the HM.

**Information provision:** The Habermas Machine does not inform participants and is only focused on finding common ground among already informed participants. Separate systems have to be built to help inform participants.

## Process Quality

### Representation

*Describe: The extent to which key decisions are representative of the constituent population.*

The Habermas Machine does not directly address the representation of participants in terms of demographic proportionality. Participants are grouped, but the system focuses on processing the content of their opinions, not ensuring a representative sample of the broader population is included in each group. The system aims to represent diverse viewpoints within the group by incorporating minority and majority opinions into the generated group statements.

*Evaluate: To what extent: (1) is there sufficient representation at critical parts of the process, including (a) proposing decisions, and (b) making ultimate decisions? (2) are there barriers leading to bias in representation?*

Sufficient representation of groups is not directly addressed through the process. As a technology-mediated system, the Habermas Machine may introduce potential barriers to entry related to technology. Lack of access to devices (computers, smartphones, tablets) and reliable internet connectivity could exclude certain segments of the population from participating. This could lead to underrepresentation of groups with lower digital literacy or limited technology access. Mitigations include building voice access, apps, or having supported access.

### Informedness

*Describe: The extent to which critical information is taken into account for decision-making.*

The Habermas Machine does not directly enhance participant informedness. It works with the opinions that participants already hold. The deliberation process itself, mediated by the AI, may indirectly increase informedness by exposing participants to diverse perspectives and prompting them

to justify and critique opinions.

*Evaluate: To what extent: (1) is critical context incorporated into decision-making about tradeoffs and consequences of different decisions? (2) is this sourced from (a) domain expertise, (b) the existing authorities, who may have extensive context, (c) a broad diversity of constituents, (d) the most impacted stakeholders, and (e) the powerful stakeholders, whose incentives are critical to having the decision "stick"?*

The HM does not actively provide participants with additional context or information about trade-offs. However, participant informedness can increase as a result of participating in the deliberative process itself—engaging with AI-generated statements that synthesize different viewpoints. Rather than direct provision of knowledge from experts, authorities, or stakeholders, this is a more indirect and emergent form of informedness. The system is designed to work with the existing knowledge and opinions of the participants, rather than to make them more informed in a traditional sense.

## Deliberation

*Describe: The extent to which decisions are considered and deliberative (rather than superficial and reactive).*

The Habermas Machine is explicitly designed to foster deliberation through an iterative process. Participants engage in multiple rounds of interaction: submitting initial opinions, reviewing and ranking AI-generated group statements, and providing critiques of the top-ranked statement. The AI mediator then uses these critiques to generate revised statements, continuing the cycle of refinement. This structured process encourages participants to consider diverse perspectives. However, it's still a very controlled form of deliberation, and no direct interaction between participants is possible.

*Evaluate: To what extent are those involved: (1) able to (and supported to) move from shallower to deeper goals and values? (2) able to (and supported to) collaborate where necessary? (3) able to address issues within the available time?*

The iterative nature of the HM process, with critique and revision rounds, encourages participants to move beyond initial, surface-level opinions. By requiring them to articulate critiques and respond to group statements, the process pushes them to consider underlying reasons and potentially evolve their perspectives towards a more nuanced understanding of the issue and shared values. While participants interact with the HM individually, the system is designed to facilitate a collective deliberation. Participants are indirectly collaborating by contributing to a shared group statement. The HM acts as a mediator to synthesize these individual

contributions into a potentially consensual output.

## Substantiveness

*Describe: The extent to which decisions are substantive (e.g., actionable, consequential) rather than nonsubstantive (e.g., vague, simplistic, inconsequential).*

The Habermas Machine aims to produce substantive outputs through group statements that capture common ground. The iterative refinement process, incorporating critiques, moves beyond vague or simplistic statements towards more comprehensive and nuanced representations of the group's collective perspective on a complex issue.

*Evaluate: To what extent: (1) is the decision directly actionable and implementable? (2) does the decision meaningfully address the issues? (3) does the decision grapple with the necessary levels of complexity? (4) is uncertainty appropriately managed and accounted for? (5) are risks to implementability accounted for?*

**Actionable**: While the group statements produced by the HM are in natural language and can be relatively detailed, they are not formal policy documents and are not directly actionable and implementable.

**Meaningfully addressing issues**: Group statements are intended to be clear and informative around issues, making use of fine-tuned LLMs to increase quality, such as in (Tessler et al., 2024).

**Complexity**: Designed for complex issues, the HM's iteration and diverse inputs allow statements to reflect multiple facets and nuances. Full complexity capture depends on input quality and LLM synthesis. However, the fact that there's no direct interaction (i.e., caucus mediation) might reduce complexity.

**Uncertainty**: HM implicitly acknowledges uncertainty by surfacing diverse views and allowing for critiques. However, in its current form, it generates a single output. A system that represents a distribution over outputs might be better at managing uncertainty.

**Risks**: As outputs are not intended to be directly implementable, the process does not have additional measures to account for risk in implementability.

## Robustness

*Describe: The extent to which the process is robust to suboptimal conditions or adversarial or strategic behavior.*

HM robustness is not fully tested, but iteration and critique may offer some resilience. Iterative critique could mitigate adversarial behavior, as statements can be challenged and refined. Still, if by posing a very extreme opinion, one might be able to influence the model more than others.

*Evaluate: To what extent is the process or system vulnerable to: (1) suboptimal conditions or broken assumptions? (e.g., low turnout, larger power asymmetries) (2) strategic behavior and manipulation? (3) false claims? (e.g., of manipulation)*

**Strategic behavior**: The Habermas Machine's effectiveness relies on the sincere input of its users, making it vulnerable to strategic manipulation if participants misrepresent their opinions to skew the outcome.

**Manipulation**: Susceptible to coordinated biased input. Limited safeguards against attacks. Transparency could offer some defense against false claims.

## Legibility

*Describe: The extent to which the processes and decisions are accessible, understandable, and verifiable.*

The Habermas Machine's process is moderately legible. The steps of opinion submission, statement generation, critique, and revision are conceptually straightforward. The final output, a group statement in natural language, is understandable. However, the inner workings of the LLM and the precise algorithms used to generate and select statements are less transparent to non-participants.

*Evaluate: To what extent is information (a) accessible, (b) understandable, (c) verifiable about the: (1) processes/systems used to make decisions? (2) the execution of these processes? (3) decisions being made (4) reasons and inputs feeding into decisions?*

**Processes/systems used to make decisions**: The reasoning behind the specific wording of the group statement is less transparent. While the intermediate statements and personalized rankings allow for inspection, the specific AI algorithms and weighting of opinions are not easily auditable by non-participants.

**Execution of processes**: Verifying the robustness of the Habermas Machine would require technical expertise to evaluate the AI models and experimental methodology. For non-participants, robustness is largely a matter of trust in the research and the described process rather than direct verification.

**Decisions being made**: The group statement, as the "decision," is presented in a clear, natural language format, making it understandable to non-participants.

**Reasons and inputs feeding into decisions**: The final group statement represents a synthesis, not a breakdown of 'for' and 'against' positions. While the HM incorporates minority opinions, the output doesn't explicitly delineate which groups favored or opposed specific aspects of the issue.

## H. Example System Card: UK Citizen Assembly on AI

The following is an example system card for a hypothetical citizen assembly run for the UK government.

## Democratic System Card

### Context

*Describe: Describe the process or a system at a high level. (Can reference a process card for more details. Can call out what is unspecified.)*

A Citizens' Assembly of 100 UK citizens chosen via sortition along geography, age, gender, education, and home ownership covariates such that they roughly match the population.

They meet for six full-day meetings and four evening meetings, totaling 60 hours of deliberation.

They go through a learning journey on AI and its place in the UK via materials and testimony from experts, stakeholders, and the unilateral authority.

They deliberate on the topic of: *How should the UK address the risks of AI persuasion?*

They collectively agree on key recommendations, creating a roadmap for enacting this plan.

The UK Government commits to publicly responding to the key recommendations.

*Describe: What are other systems that this process depends on or interacts with, which impact its success? (e.g., sortition data, or user or citizen authentication systems)*

The organizers require access to relevant data to carry out the sortition effectively and in a representative manner.

A wider communications campaign also complements the process to support recruitment, awareness raising, and opportunities for wider input. The process generally relies on a wider engagement program that supports the constituent population to contribute and participate in the process beyond the membership of the citizens' assembly, such as through contributing their values, views, desired outcomes and concerns in a structured and representative manner.

### Process Quality

#### Representation

*Describe: The extent to which key decisions are representative of the constituent population.*

Members are selected via sortition to be a demographically proportional representation of the UK public by age, gender, geography, education, and home ownership.

*Evaluate: To what extent: (1) is there sufficient representation at critical parts of the process, including (a) proposing decisions, and (b) making ultimate decisions? (2) are there barriers leading to bias in representation?*

Assembly members are fully involved in making final recommendations. However, the agenda-setting and scoping are initially structured by the organizing team, introducing some limits to representation at the scoping stage. As the process progresses, assembly members gain more agency in proposing new decision options and directions, but their efficacy is ultimately constrained by conditions imposed by the UK government.

Some recruitment processes may miss people from some demographics or not adequately control for groups beyond stratification covariates. Some groups typically engage less with political processes due to a myriad of factors, so additional work is necessary to guarantee the necessary engagement from this public. Only 100 people are chosen, so many subgroups and their intersections miss out on membership. The assembly members make the key decisions (which are reflective of the views of the assembly and not necessarily the constituent population). Some of these gaps are addressed in information provision, but not all.

#### Informedness

*Describe: The extent to which critical information is taken into account for decision-making.*

Assembly members are taken on a learning journey, including engagement with diverse information, hearing from a variety of experts in AI development, industrial policy, AI governance, and public service innovation, the views of key stakeholders, and the lived experiences of the other assembly members.

*Evaluate: To what extent: (1) is critical context incorporated into decision-making about tradeoffs and consequences of different decisions? (2) is this sourced from (a) domain expertise, (b) the existing authorities, who may have extensive context, (c) a broad diversity of constituents, (d) the most impacted stakeholders, and (e) the powerful stakeholders, whose incentives are critical to having the decision "stick"?*

Extended time allows for an in-depth learning journey that provides a baseline understanding of context and the tradeoffs between considerations.

Opportunities for unilateral authority and stakeholder feedback on draft recommendations facilitate understanding of impacts and trade-offs.

Current practices may not accommodate diverse learning styles and participants' differing ability to digest large amounts of written or oral information, resulting in some uneven understanding of the issue. Capabilities such as scenario mapping and impact forecasting are technically and practically constrained by time.

### Deliberation

*Describe: The extent to which decisions are considered and deliberative (rather than superficial and reactive).*

Citizens spend 60 hours deliberating with each other, the process is managed by independent facilitators, and the process makes use of mixed breakout groups, plenary sessions and other discussion formats.

*Evaluate: To what extent are those involved: (1) able to (and supported to) move from shallower to deeper goals and values? (2) able to (and supported to) collaborate where necessary? (3) able to address issues within the available time?*

Independent facilitation provides structured formats for assembly members to develop their views and reconcile them with the views of others through conversation and group work.

The 60 hours of deliberation provide sufficient time for addressing core issues within the remit, although some assembly members always report feeling pressed for time when tackling particularly complex aspects of AI governance. Facilitators helped manage the workflow to ensure all critical decision-making steps were met, key issues received adequate attention, and the process concluded with results.

### Substantiveness

*Describe: The extent to which decisions are substantive (e.g., actionable, consequential) rather than nonsubstantive (e.g., vague, simplistic, inconsequential).*

A carefully facilitated process ensures that recommendations respond to the remit and consider the key problems presented to the assembly, with purposeful attention paid to the systems that their recommendations will be interacting with to optimally design for implementability.

*Evaluate: To what extent: (1) is the decision directly actionable and implementable? (2) does the decision meaningfully address the issues? (3) does the decision grapple with the necessary levels of complexity? (4) is uncertainty appropriately managed and accounted for? (5) are risks to implementability accounted for?*

Final recommendations respond directly to the remit and address values-laden social trade-offs. The outputs are clear in their intent and demonstrate an understanding of relevant uncertainty. The facilitation process ensured that final recommendations were concrete and actionable rather than settling for superficial agreement.

Throughout the process, experts and policymakers provided input, helping assembly members account for potential implementation challenges and barriers.

The final outputs are limited in their thoroughness due to practical constraints and so require interpretation during implementation by policymakers.

### Robustness

*Describe: The extent to which the process is robust to suboptimal conditions or adversarial or strategic behavior.*

The sortition process is exposed to some manipulation risks due to demographic reporting and quota settings but informational processes and group decision-making were robust due to clear rules and standards.

*Evaluate: To what extent is the process or system vulnerable to: (1) suboptimal conditions or broken assumptions? (e.g., low turnout, larger power asymmetries) (2) strategic behavior and manipulation? (3) false claims? (e.g., of manipulation)*

Low turnout would have broken participant recruitment. Recruitment processes were subject to possible manipulation strategies due to the selection process. Transparency, governance integrity, and diverse stakeholder buy-in defeat false claims.

### Legibility

*Describe: The extent to which the processes and decisions are accessible, understandable, and verifiable.*

Recommendations are made public. Templated outputs generally require explanatory reasoning. The process is open to observers and scrutineers. Open public communications pre-output pre-empt partisan distrust.

*Evaluate: To what extent is information (a) accessible, (b) understandable, (c) verifiable about the: (1) processes/systems used to make decisions? (2) the execution of these processes? (3) decisions being made (4) reasons and inputs feeding into decisions?*

All results were made public and data was opened to outside review. However, due to the detailed nature of the work and sheer volume of the data used, not every in-depth element was legible to all outside parties. All captured datapoints were made accessible through a searchable database. Identifiable participant voting records are not made public.

## Delegation

### Integration

*Describe: The extent to which the authority integrates the democratic process into its operations.*

The UK government specifically focused the remit on areas where they had the capability to integrate this process directly into existing and future decision-making around the development of an action plan.

*Evaluate: To what extent is the authority structuring its internal communications and operations to effectively: (1) provide critical context to democratic process / system? (2) integrate democratic process outputs in its actions? (3) trigger democratic processes when/if required?*

The decisions themselves cannot be automatically implemented due to their long-term strategic nature, which also leaves them open to adjustment by the unilateral authority.

Departmental staff were directly involved in the process through observation and feedback phases, enriching their understanding of the intent of final recommendations and their overall ability to infer preferences when faced with implementation gaps.

### Ability to bind

*Describe: The extent to which the authority is able to technically and legally bind itself to democratic decisions.*

The UK government has agreed to present the resulting recommendations to parliament, but the process is not meant to be binding.

*Evaluate: To what extent can the unilateral authority bind itself to acting in accordance with the democratic decision: (1) technically? (2) legally? (e.g., has developed the needed technical and/or legal infrastructure for binding)*

In theory, there are ways that assembly decisions could be made binding by Parliament, but that is out of scope for this particular process.

## Commitment

*Describe: The extent to which the unilateral authority commits to acting in accordance with the democratic decision.*

The UK government pre-committed to publicly responding to the final recommendations and implementing them to the maximum extent possible (conditional on acceptance of the recommendation in principle).

*Evaluate: To what extent has the unilateral authority committed to acting in accordance with the democratic decision: (1) internally? (2) privately? (3) publicly? (regardless of their ability to bind)*

There is a verbal commitment to enact the recommendation to the maximum extent possible, although the extent to which this is true is largely up to the unilateral authority and will not be clear for a number of years.

## Trust

### Awareness

*Describe: The extent to which the relevant public is aware of the democratic process.*

The UK public was made aware of the process through a public communications campaign that complemented the process.

*Evaluate: To what extent is the relevant public aware: (1) that the democratic system exists? (2) how it works? (3) what it is being used for? (4) how they can be involved?*

The public has a low level of awareness. There is some coverage in specialised media and a broader public communications campaign, including advertising pathways to be included, but there is generally little public engagement with government policy-making on this topic.

Public communications clearly explain that the assembly would inform UK policy on AI persuasion risks, though a detailed understanding of what exactly this process would look like or how exactly the recommendations would influence policy development was limited among the general public.

### Participation

*Describe: The extent to which the relevant public is willing to participate in the process.*

Response rates to recruitment invitations are in line with global averages but lower than the most successful examples in neighbouring Ireland.

*Evaluate:* *To what extent is the relevant public: (1) willing to participate? (2) able to participate? (3) appropriately compensated for participating? (4) actually participating?*

There was a sufficient pool of the public eager to participate. They were generously reimbursed for their time.

## Accountability

*Describe:* *The extent to which there are external watchdogs and accountability structures monitoring the execution of the democratic process and the implementation of its outputs.*

An independent governance body is established to oversee the process and hold the UK government to account by reporting on recommendation implementation progress.

*Evaluate:* *To what extent are: (1) there well-understood lines of oversight and accountability? (2) sufficiently influential/powerful organizations focused on holding authorities to their promised levels of democratic involvement? (3) authorities and democratic systems responsive to such accountability mechanisms?*

The independent governance body had limited powers to mandate accountability, but its public profile commanded responsive actions where needed.

## Buy-in

*Describe:* *The extent to which the relevant public and key stakeholders buy into the process and its legitimacy.*

Key industry stakeholders, civil servants responsible for implementing recommendations, and political leaders are included in the process planning stage to establish commitments before outputs are generated.

*Evaluate:* *To what extent are the relevant public and key stakeholders accepting of the legitimacy of: (1) the system/process? (2) of the decision?*

Their involvement in the process implicates them in building legitimacy. When compared to existing policy-making and political decision-making processes the process is viewed as considered and reasonable because of its resistance to shallow public opinion and electoral incentives.

# I. Additional examples of Democratic System Cards

See democracylevels.org/system-card for examples of democratic system cards for real-world processes and systems. As a continuation of this body of work, this resource will include tools and resources that can support authorities and stakeholders in applying system cards for the implementation and evaluation of democratic systems.

