# OpenReview forum: "Position: Democratic AI is Possible. The Democracy Levels Framework Shows How It Might Work."
_ICML.cc/2025/Position_Paper_Track — ICML 2025 Position Paper Track poster_

### Official Review · Reviewer_sca5 · 2025-03-14

**Significance:** 1
**Argument Clarity:** 3
**Rating:** 2
**Confidence:** 4

**Questions:**

What evidence supports the claim that democratic AI governance is feasible at scale? Imagine every person on the world will be somehow using AI for this -- thats a crazy amout of resources used to provide such a model.

How does the framework handle conflicts between democratic decisions and commercial interests?

**Discussion Potential:**

3

**Paper Summary:**

The paper argues that democratizing AI governance is necessary to ensure public legitimacy and accountability. It introduces the Democracy Levels Framework, which outlines a staged approach to integrating democratic processes into AI decision-making. The framework ranges from unilateral corporate control (L0) to fully democratic constitutional structures (L5). The authors cite existing AI governance experiments, such as Meta’s Community Forums and Anthropic’s Collective Constitutional AI, as early steps toward democratic AI.

While the paper presents a normative case for democratic AI, it largely assumes that more democracy in AI governance is beneficial, without deeply addressing practical limitations or trade-offs. It engages with alternative perspectives, such as corporate control models and accelerationist critique, but does not critically assess the feasibility of democratic AI governance at scale.

**Position:**

Yes

**Position In Title:**

Yes

**Related Work:**

2

**Strengths And Weaknesses:**

The strenght of the paper is its structured framework and conceptual clarity. It is offering a roadmap for organizations which aim for a public legitimacy in AI governance. The discussion of existing corporate AI governance efforts provides a bigger context. The introduction of practical assessment tools adds beyond theoretical discussion.

However, the paper lacks empirical validation and remains highly idealistic. The Democracy Levels Framework is a rigid taxonomy rather than a practical implementation guide, offering no concrete steps for organizations transitioning between levels. The authors fail to engage with real-world constraints. One can imagine stuff like corporate incentives or a public path public towards AI governance issues. The assumption that AI-driven deliberation could enhance democratic decision-making is not clear. It is ignoring risks such as algorithmic bias which can shap the public discourse. Moreover, the paper neglects the expertise gap in AI governance. Many AI decisions require technical knowledge, yet the framework does not reconcile public participation with expert oversight -- older people? Without mechanisms to balance technical constraints and democratic legitimacy, the framework risks oversimplifying governance challenges and becoming infeasible in practice.

**Support:**

2

---

> ### Author Rebuttal · Authors · 2025-03-31
>
> Thanks for your thoughtful feedback! Overall we want to emphasise that according to the call, position papers “make an argument for a viewpoint or perspective about what should be done.” While of course it is appropriate to cite existing evidence that supports or challenges the position, the primary purpose of a position paper is not to “provide empirical validation,” and we think there needs to be room for arguments for directions that there are normative reasons to pursue, even with significant uncertainty about to what extent this approach can be practically implemented. In part, the purpose of this paper is to spur further work that will help answer this question.
>
> Your questions:
>
> - “What evidence supports the claim that democratic AI … is feasible at scale? …”
>     - We take the point that this could be resource intensive, but there are many existing examples from other domains, including the EU’s Citizens’ Panels, permanent citizens assemblies, Meta’s Community Forums, widespread use of participatory budgeting, etc. It’s plausible that this could be scaled up. Proposed models like ARIA’s Safeguarded AI or case-based reasoning would require a manageable level of resourcing. It’s possible (but not guaranteed) that use of AI could help make conducting such processes at scale less resource-intensive. The Decision Tool is intended to help navigate trade-offs between democracy + costs.
>
> - “How does the framework handle conflicts between democratic decisions and commercial interests?”
>     - The framework is intended to help articulate and navigate trade-offs between commercial interests and democratic ideals. For example, the Levels Decision Tool can help a company assess how much it may need to defer to democratic processes, based on how much of a legitimacy problem it has for a given decision. If the decision made by the democratic process is not that which is profit-maximizing, the company might be faced with a choice between the (extra) profit, and the legitimacy / goodwill of adhering to the democratic decision. In other work, we are exploring legal structures which would allow companies to bind themselves to the outputs of democratic processes for certain decisions. The framework can also apply to non-corporate entities (e.g., governments/regulators, not-for-profit labs).
>
> Other points:
>
> - “... without deeply addressing practical limitations or trade-offs” etc
>     - Respectfully, we want to push back on these points: the whole framework is designed around the assumption that there are real world constraints and trade-offs. The Democratic System Card is designed to help understand the ways in which democratic processes are imperfect (e.g., because of real-world constraints), and the Levels Decision Tool is explicitly for navigating the trade-offs associated with different levels of democratic-ness. We will make this more explicit in the text, and add another alternative view that speaks to skepticism that this approach is feasible.
>
> - “one can imagine … corporate incentives or a public path towards AI governance”
>     - We already include the ‘government-only critique’ and make the case in 2.3 that use of democratic systems for decisions that have a high need for legitimacy can be consistent corporate incentives. What else would you like to see?
>
> - “The … Framework is a rigid taxonomy rather than a practical implementation guide, offering no concrete steps for … transitioning between levels”
>     - We see the three artifacts we provided (the level definitions, the Levels Decision Tool, and the Democratic System Card) as prerequisites to providing a well-motivated set of concrete recommendations. We agree that concrete advice of this sort is important, and are actively developing such guidance for future work, building on the foundation provided by this position paper.
>
> - “The assumption that AI-driven deliberation could enhance democratic decision-making is not clear. It is ignoring risks such as algorithmic bias …”
>     - We agree that there are serious risks to incorporating AI and will update the paper to better acknowledge these. Our genuine stance is that it’s merely possible that AI could be used in ways that enhance democratic decision-making, so long as such use cases are explored with appropriate care and while being clear-eyed such risks. At least, tools like LLMs expand the space of decision processes that are possible, and there is already work exploring how they can be used responsibly (e.g., with extensive human-in-the-loop as in Konya et al 2025).
>
> - “the paper neglects the expertise gap in AI governance … older people?”
>     - The expertise gap is of course a challenge, but there are ways to approach it, and this is true of many domains in which there have been (e.g.) successful democratic processes such as citizens assemblies, which have mechanisms to deal with it (education components, expert consultation, etc) (Warren & Gastil, 2015; Leino et al., 2022).

---

### Official Review · Reviewer_Uhp4 · 2025-03-16

**Significance:** 3
**Argument Clarity:** 2
**Rating:** 3
**Confidence:** 3

**Questions:**

See above

**Discussion Potential:**

3

**Paper Summary:**

This work makes the case for "democratic AI" and proposes a corresponding "democracy levels framework" that tracks various components of what makes a system "democratic." The authors define democratic AI as "systems for collective decision-making that are 'characterized by[] equality among participants' that influence the development, alignment and governance of AI." The corresponding "levels" of democracy in their framework are about the extent to which 'democratic' processes affect decision-making about AI governance policies.

**Position:**

Yes

**Position In Title:**

Yes

**Related Work:**

1

**Strengths And Weaknesses:**

The paper is overall well written, thoughtful, and a lot of work was clearly put into thinking through examples for this paper. The discussion of alternative views, in particular, is nice in the sense that the presented alternatives are realistic (rather than straw-person) counterpoints.

However, I think there are some ambiguities about core concepts that get somewhat swept under the rug. I want to emphasize that my comments here are not meant to reflect my disagreement with the position or an example of a conversation that might be generated by the position -- I feel like these are issues that actually hinder a hypothetical discussion of what we might actually want out of 'democratic' AI.

* Democracy as a concept vs buzzword. Why is _democracy_ specifically important? I understand the phrase "democratic AI" is thrown around left and right these days, but it feels odd to claim the term while essentially not engaging literally centuries of political theory on it, how it is defined, what properties it has, how to implement it, various benefits and limitations (e.g. you may be interested in Estlund as a citation for epistemic benefits of democracy).
* AI specificity. If 'democracy' as this paper understands it is generally about "how the public can impact rules that govern a system", why is something AI- specific necessary? The levels in the given framework --- and their examples --- are merely about the extent to which the central decisionmaker incorporates public input. Is there any reason your framework doesn't also apply to (e.g.) HCI-type work that studies online communities and self-governance, or organizational sociology work on (e.g.) different types of corporate or coop structures?
* What kinds of "decisions" count? The way the framework and its examples are written make it seem as though "democratic AI" is _only_ about the rules that govern it (i.e. meta-level rules, e.g. policy on persuasion) even though lots of the other work that uses the "democratic AI" tag is specifically about inputs to the learning process itself (e.g. learning from heterogeneous data or pluralistic alignment).

Perhaps what I'm getting at here is this: If your actual position is something like _When we use the word "Democratic AI", we should be talking about the extent to which the public has input into its rules for governance_, that feels like a stronger position that can actually lend itself to vigorous debate. But as written, I don't feel that the paper actually supports _Democratic AI is possible and valuable_.

**Support:**

3

---

> ### Author Rebuttal · Authors · 2025-04-01
>
> Thanks for your thoughtful engagement with our work!
>
> 1. (re. **justifications for democracy + connections to literature**) We think we’re on the same page here — section 2.3 was our attempt to summarize the literature for an ICML audience as best we could within the length constraints, but we can do it better. Our framework centers on decision-making power, so we’ll update the motivation to summarize the political philosophy literature that justifies democracy as a way to (i) legitimize the use of power (e.g., via consent, contestability, open deliberation, etc; per Locke, Jefferson, Habermas), (ii) distribute power in ways that reflect intrinsic moral equality  (e.g., Dahl, Buchanan) and prevent domination (e.g., Pettit), and (iii) achieve epistemic benefits that are more difficult when decision-making power is centralized (e.g., Estlund, Landemore).
>
> 2. (re. **AI specificity**) The levels are definitely more general and could be applied in many different contexts,, and we may well publish a later version of the framework in a more general venue. But we believe that of the many contexts in which it could be applied, AI is particularly urgent given the risks of power concentration this technology might pose. E.g., AI development and deployment can be resource intensive in ways that naturally concentrate infrastructural power, and the fact that a small number of companies provide frontier models which are very widely used means that for the most part the public has no direct influence over systems which they are still affected by. Thus, there is a critical need to articulate these ideas for the ICML audience, who are one of the key constituencies in a position to influence the trajectory of this technology. We’re happy to call this out the more general applicability explicitly in the paper, but don’t think it undermines the work. To our knowledge, even among more general frameworks for evaluating democracy (which we cite in 3.5), there is nothing quite like the framework we provide.
>
> 3. (re. **What kinds of decisions count?**) The framework is meant to be agnostic on the question of which decisions should be made democratically. We give examples that we think are strong candidates in 5.1, and don’t intend to limit to ‘meta-level rules’ — decisions about inputs to the learning process are also in scope (e.g., pluralistic alignment methods could be evaluated against the levels and dimensions as a process for determining model behavior, though may score lower on dimensions like deliberativeness). For many decisions democracy is not that important (e.g., because they have negligible impact beyond the decision-maker), and democracy itself has costs (in speed, attention, labour etc) that must be traded off against benefits such as improved legitimacy. The Levels Decision Tool is designed to be used to help make this determination (and we can clarify this in the paper). Generally we don’t think the main contribution of this paper is to be highly prescriptive in this respect. Any such determinations will be contested, so we aim simply to give enough examples of decisions (that enough readers will agree should not be made unilaterally) to convey that this direction is worth pursuing, at least for some important decisions. We are working on other artifacts (developed through broad consultation) that aim to be more prescriptive about what decisions are in scope.
>
> It seems you have overall reservations that might not be addressed above, so to make sure we address those:
>
> 4. You say that there are “ambiguities about core concepts that get … swept under the rug” and that these “hinder a hypothetical discussion of what we might actually want out of 'democratic' AI”. Given your bullets, we assume you are referring to ambiguity about why (1) democracy is valuable, (2) why we are using this framework for AI specifically, and (3) what decisions count. Do the above responses sufficiently address these?
>
> 5. You say “I don't feel that the paper actually supports Democratic AI is possible and valuable.”, and suggest a better position statement would focus on the meaning of “Democratic AI”. If we better make the case that democracy is valuable (with better links to existing literature as described above) and possible (with better citing of existing examples of democratic processes that operate at scale, see response to Reviewer 4), does that address your concerns? We would rather stay with the ‘possible + valuable’ position (and simply state the definition of ‘democratic AI’ we are using), rather than change to a position on what the phrase ‘democratic AI’ should mean, which as a semantic issue seems less substantive. Please also consider that what it means for democratic AI (as we define it) to be valuable will be different for different audiences (because people value democracy for pluralistic reasons), and that with this paper we are trying to make the case to diverse audiences, including companies and AI researchers.

---

> > ### Comment · Reviewer_Uhp4 · 2025-04-04
> >
> > Thanks for the response - these are useful clarifications that I will be happy to see in revisions.
> >
> > I will update my score but I do want to say: defining "democratic AI" is itself a position! (Maybe it's semantic but semantics _can be_ substantive etc etc....) And the paper is implicitly taking a position on how it _ought to_ be implemented by choosing to focus on a particular framework set up in a particular way.

---

### Official Review · Reviewer_RYZ5 · 2025-03-16

**Significance:** 4
**Argument Clarity:** 4
**Rating:** 4
**Confidence:** 4

**Questions:**

This may be implemented in each country or coalition, but how can this be implemented globally, by prohibiting or making it disadvantageous to defect.
The present arguments seem to focus on alignment of individual AI agents by human users/developers/regulators, but isn't it also important to implement the process of democracy among AI agents so that power concentration to particular AI agent is avoided?
The word "persuasion" suddenly appears in Figure 2. Could you explain the meaning in the preceding text?
Figure 2 was partly occluded in my Preview on Mac. Could this be fixed?

**Discussion Potential:**

4

**Paper Summary:**

This paper argues that democracy in AI development and deployment is important and possible. The authors defines Five levels of AI democracy and proposes the System Cards and Decision Tool for implementation and evaluation of AI democracy.

**Position:**

Yes

**Position In Title:**

Yes

**Related Work:**

3

**Strengths And Weaknesses:**

The position of the authors are clearly presented. They also try to strengthen their position by providing tools of assessment and implementation of AI democracy. A good Q&A is included.

**Support:**

3

---

> ### Author Rebuttal · Authors · 2025-03-31
>
> - (re. **global implementation**) Thanks for asking about global implementation. While the framework is designed to be agnostic about the geographical scale at which it is implemented, we certainly see the value of global coordination to address some of the threats AI poses. That said, there is an extensive and growing literature on the design of (and prospects for) global governance of AI, and we don’t think we can meaningfully cover it in this paper. That said, some of the authors see  significant opportunity for some of the democratic mechanisms alluded to here  for global implementation, with game-theoretic reasons why this might be effective, but that would be the subject of a future paper.
>
> - (re. **democracy over agents**) This is a great point! We can imagine situations in which democratic approaches might be useful for mitigating power concentration among AI agents, and we will flag this in the paper. That said, our primary focus in this paper is on aligning with people. For example, we can imagine a democracy over AI agents, none of which are particularly aligned with human interests; or conversely a hypothetical where a powerful single agent is nonetheless approximately aligned with the democratically chosen will of humanity (respecting pluralism, subsidiarity etc).
>
> - (re. **persuasion**) We did try to explain our use of the word persuasion on line 217 when we introduce Figure 2, but we can make this clearer. We are referring to the phenomena where, e.g., a chatbot can persuade people and change their views ([Costello et al., 2024](https://www.science.org/doi/abs/10.1126/science.adq1814); [Matz et al., 2024](https://www.nature.com/articles/s41598-024-53755-0)), which there are lots of ethical issues with ([Bezou-Vrakatseli et al., 2023](https://arxiv.org/pdf/2309.04352)).
>
> - We’ll fix the **issue with Figure 2** being occluded on Mac — this an intermittent bug with Figma exports (and Mac CPU hardware acceleration in some viewers) that we are aware of now and will fix.

---

> > ### Comment · Reviewer_RYZ5 · 2025-04-05
> >
> > I look forward to seeing your revised and future publications.

---

### Official Review · Reviewer_sbij · 2025-03-19

**Significance:** 3
**Argument Clarity:** 4
**Rating:** 4
**Confidence:** 4

**Questions:**

1. Another dimension to consider might be the degree to which democratic representation is direct. Do individual users provide input to the democratic process? Or do they elect representatives who provide direct input?

2. Another role to consider might be the extent to which a democratic process has control over the particular way in which it democratically aggregates inputs. E.g. suppose the democratic process is a voting rule, e.g. Borda. Does the authority allow the democratic process to choose a different voting rule, e.g. random dictator? (Similarly, we might ask whether a democratic process is given the authority to decide how to adjust its constituency.)

3. Do the authors have thoughts on when democracy is valuable and when it isn’t? (e.g. perhaps in many contexts, deeply wrong and oppressive policies should not be put to a democratic vote) Or is this just a framework for reasoning about how democratic a decision-making process is, which does not imply anything about how good it is? The evaluations produced by the democratic system card are helpful in this respect, but I think the system card could be clearer about (a) which criteria are helping us to assess whether a democratic (or some alternative) process should be used, (b) which ones are helping us to assess whether a process is democratic, and (c) which ones are helping us to assess whether the process has certain generally good features, which have nothing especially to do with democracy, e.g. robustness. The discussion in 5.1 is also helpful, but more examples in the bulleted list, as well as some references to current philosophy literature on the value of democracy (e.g. from the SEP on democracy) would be helpful.

**Discussion Potential:**

3

**Paper Summary:**

The authors propose a framework for assessing the extent to which a system is democratic. This includes (a) a description of the roles that democratic processes can play in the pipeline from AI regulation, to AI organizations, to AI systems; (b) a description of the different roles that can be performed by democratic processes, as well as a set of questions (the Levels Decision Tool) to guide reflection on which role to assign in a context; (c) a Democratic System Card, a set of questions to guide reflection on whether the quality of a democratic system is commensurate with the level of decision-making power delegated to it.

## update after rebuttal

I thank the authors' for their reply. I maintain my score.

**Position:**

Yes

**Position In Title:**

Yes

**Related Work:**

4

**Strengths And Weaknesses:**

This is a very clear paper, and the framework it proposes is informative and practical. It is well argued and cites relevant literature. It seems relevant to the ICML community and likely to inspire discussion. I discuss some possible areas for revision in the questions below.

**Support:**

4

---

> ### Author Rebuttal · Authors · 2025-03-31
>
> 1. Thanks for the suggestion — it’s an important distinction but as much as possible we try to keep the framework agnostic to the structure of the democratic system/process, because we don’t think there is any consensus on whether, e.g., direct democracy is necessarily more ‘democratic’ than representative democracy. Both can in principle satisfy the desiderata captured by the dimensions, depending on how they are implemented.
> 2. This kind of metagovernance is what we have in mind for L5 in the framework (indeed, we originally called the decision remit that is added at L4 ‘metagovernance’, but changed it to ‘constitutional structures’ as we thought it was a bit more concrete). We’ll make sure to clarify this!
> 3. Thanks for this thoughtful feedback!
>     - Our intention is for the framework to capture how democratic a decision-making process/system is, regardless of how desirable democracy is for making any given decision. Of course, we have views on when democracy is desirable, and section 5.1 was our attempt to be transparent about this, while acknowledging that such normative prescriptions are contested. We’ll add some more examples to the bulleted list and better connect the motivation to the philosophy literature on justifications for democracy (for more on this, see our response to Reviewer 3 / “Uhp4” below).
>     - Thanks for the suggested (a) / (b) / (c) breakdown of the dimensions in the System Card. We iterated a lot on how to factor the space of concepts and while this breakdown does seem natural, we don’t think it quite works because there are many interactions between the dimensions (e.g., if a process is not robust to low turnout or strategic behavior, the decisions it makes may be less representative), and whether a given democratic process should be used is a determination that depends not only on the (normative) value of democracy in that context but also on the resource constraints and other trade-offs (captured by the Levels Decision Tool), the quality of the processes available, and the degree to which other conditions (e.g, commitment of the unilateral authority) are such that the process can deliver on the democratic value sought. So e.g., (a) depends on (b) and (c). That all said, we suspect many will wonder why we don’t factor things in this way so we’ll clarify this in the camera-ready version.

---

### Decision · Program_Chairs · 2025-04-29

**Decision:**

Accept (poster)

**Comment:**

Overall, 3 out of 4 reviewers are enthusiastic about this position paper, praising its clarity, informativeness, practicality as well as relevance to the ICML community. On the other hand, one reviewer has some reservations on the practicality of the proposed framework, saying the proposed position is highly idealistic. The authors are used to seriously address this major criticism in their revision of the paper.  They should also remove the "chatgpt" element from the Brookings Institution reference URL.  (Use of ChatGPT to generate references is extremely risky; authors should carefully check every reference.)